# Deep Convolutional Inverse Graphics Network

**Tejas D. Kulkarni\*[1], William F. Whitney\*[2],**
**Pushmeet Kohli[3], Joshua B. Tenenbaum[4]**
[1,2,4]Massachusetts Institute of Technology, Cambridge, USA
[3]Microsoft Research, Cambridge, UK
[1]tejask@mit.edu  [2]wwhitney@mit.edu  [3]pkohli@microsoft.com  [4]jbt@mit.edu
*\* First two authors contributed equally and are listed alphabetically.*

## Abstract

This paper presents the Deep Convolution Inverse Graphics Network (DC-IGN), a model that aims to learn an interpretable representation of images, disentangled with respect to three-dimensional scene structure and viewing transformations such as depth rotations and lighting variations. The DC-IGN model is composed of multiple layers of convolution and de-convolution operators and is trained using the Stochastic Gradient Variational Bayes (SGVB) algorithm [10]. We propose a training procedure to encourage neurons in the *graphics code* layer to represent a specific transformation (e.g. pose or light). Given a single input image, our model can generate new images of the same object with variations in pose and lighting. We present qualitative and quantitative tests of the model's efficacy at learning a 3D rendering engine for varied object classes including faces and chairs.

## 1   Introduction

Deep learning has led to remarkable breakthroughs in learning hierarchical representations from images. Models such as Convolutional Neural Networks (CNNs) [13], Restricted Boltzmann Machines, [8, 19], and Auto-encoders [2, 23] have been successfully applied to produce multiple layers of increasingly abstract visual representations. However, there is relatively little work on characterizing the optimal representation of the data. While Cohen *et al.* [4] have considered this problem by proposing a theoretical framework to learn irreducible representations with both invariances and equivariances, coming up with the best representation for any given task is an open question.

Various work [3, 4, 7] has been done on the theory and practice of representation learning, and from this work a consistent set of desiderata for representations has emerged: invariance, interpretability, abstraction, and disentanglement. In particular, Bengio *et al.* [3] propose that a *disentangled* representation is one for which changes in the encoded data are sparse over real-world transformations; that is, changes in only a few latents at a time should be able to represent sequences which are likely to happen in the real world.

The "vision as inverse graphics" paradigm suggests a representation for images which provides these features. Computer graphics consists of a function to go from compact descriptions of scenes (the *graphics code*) to images, and this graphics code is typically disentangled to allow for rendering scenes with fine-grained control over transformations such as object location, pose, lighting, texture, and shape. This encoding is designed to easily and interpretably represent sequences of real data so that common transformations may be compactly represented in software code; this criterion is conceptually identical to that of Bengio *et al.*, and graphics codes conveniently align with the properties of an ideal representation.

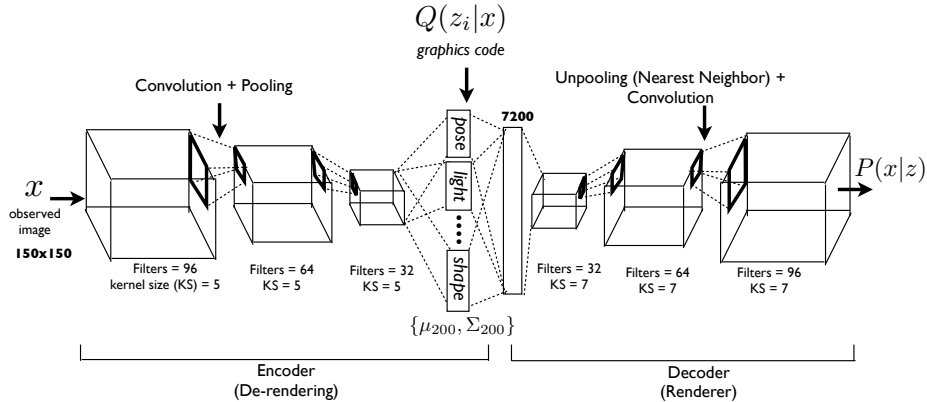

Figure 1: **Model Architecture:** Deep Convolutional Inverse Graphics Network (DC-IGN) has an encoder and a decoder. We follow the variational autoencoder [10] architecture with variations. The encoder consists of several layers of convolutions followed by max-pooling and the decoder has several layers of unpooling (upsampling using nearest neighbors) followed by convolution. (a) During training, data $x$ is passed through the encoder to produce the posterior approximation $Q(z_i|x)$, where $z_i$ consists of scene latent variables such as pose, light, texture or shape. In order to learn parameters in DC-IGN, gradients are back-propagated using stochastic gradient descent using the following variational object function: $-log(P(x|z_i)) + KL(Q(z_i|x)||P(z_i))$ for every $z_i$. We can force DC-IGN to learn a disentangled representation by showing mini-batches with a set of inactive and active transformations (e.g. face rotating, light sweeping in some direction etc). (b) During test, data $x$ can be passed through the encoder to get latents $z_i$. Images can be re-rendered to different viewpoints, lighting conditions, shape variations, etc by setting the appropriate graphics code group ($z_i$), which is how one would manipulate an off-the-shelf 3D graphics engine.

Recent work in inverse graphics [15, 12, 11] follows a general strategy of defining a probabilistic with latent parameters, then using an inference algorithm to find the most appropriate set of latent parameters given the observations. Recently, Tieleman *et al.* [21] moved beyond this two-stage pipeline by using a generic encoder network and a domain-specific decoder network to approximate a 2D rendering function. However, none of these approaches have been shown to automatically produce a semantically-interpretable graphics code and to learn a 3D rendering engine to reproduce images.

In this paper, we present an approach which attempts to learn interpretable *graphics codes* for complex transformations such as out-of-plane rotations and lighting variations. Given a set of images, we use a hybrid encoder-decoder model to learn a representation that is disentangled with respect to various transformations such as object out-of-plane rotations and lighting variations. We employ a deep directed graphical model with many layers of convolution and de-convolution operators that is trained using the Stochastic Gradient Variational Bayes (SGVB) algorithm [10].

We propose a training procedure to encourage each group of neurons in the *graphics code* layer to distinctly represent a specific transformation. To learn a disentangled representation, we train using data where each mini-batch has a set of active and inactive transformations, but we do not provide target values as in supervised learning; the objective function remains reconstruction quality. For example, a nodding face would have the 3D elevation transformation active but its shape, texture and other transformations would be inactive. We exploit this type of training data to force chosen neurons in the *graphics code* layer to specifically represent active transformations, thereby automatically creating a disentangled representation. Given a single face image, our model can re-generate the input image with a different pose and lighting. We present qualitative and quantitative results of the model's efficacy at learning a 3D rendering engine.

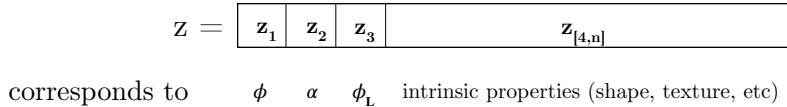

corresponds to $\phi$ $\alpha$ $\phi_L$ intrinsic properties (shape, texture, etc)

Figure 2: **Structure of the representation vector.** $\phi$ is the azimuth of the face, $\alpha$ is the elevation of the face with respect to the camera, and $\phi_L$ is the azimuth of the light source.

## 2 Related Work

As mentioned previously, a number of generative models have been proposed in the literature to obtain abstract visual representations. Unlike most RBM-based models [8, 19, 14], our approach is trained using back-propagation with objective function consisting of data reconstruction and the variational bound.

Relatively recently, Kingma *et al.* [10] proposed the SGVB algorithm to learn generative models with continuous latent variables. In this work, a feed-forward neural network (encoder) is used to approximate the posterior distribution and a decoder network serves to enable stochastic reconstruction of observations. In order to handle fine-grained geometry of faces, we work with relatively large scale images ($150 \times 150$ pixels). Our approach extends and applies the SGVB algorithm to jointly train and utilize many layers of convolution and de-convolution operators for the encoder and decoder network respectively. The decoder network is a function that transform a compact *graphics code* ( 200 dimensions) to a $150 \times 150$ image. We propose using unpooling (nearest neighbor sampling) followed by convolution to handle the massive increase in dimensionality with a manageable number of parameters.

Recently, [6] proposed using CNNs to generate images given object-specific parameters in a supervised setting. As their approach requires ground-truth labels for the *graphics code* layer, it cannot be directly applied to image interpretation tasks. Our work is similar to Ranzato *et al.* [18], whose work was amongst the first to use a generic encoder-decoder architecture for feature learning. However, in comparison to our proposal their model was trained layer-wise, the intermediate representations were not disentangled like a *graphics code*, and their approach does not use the variational auto-encoder loss to approximate the posterior distribution. Our work is also similar in spirit to [20], but in comparison our model does not assume a Lambertian reflectance model and implicitly constructs the 3D representations. Another piece of related work is Desjardins *et al.* [5], who used a spike and slab prior to factorize representations in a generative deep network.

In comparison to existing approaches, it is important to note that our encoder network produces the interpretable and disentangled representations necessary to learn a meaningful 3D graphics engine. A number of inverse-graphics inspired methods have recently been proposed in the literature [15]. However, most such methods rely on hand-crafted rendering engines. The exception to this is work by Hinton *et al.* [9] and Tieleman [21] on *transforming autoencoders* which use a domain-specific decoder to reconstruct input images. Our work is similar in spirit to these works but has some key differences: (a) It uses a very generic convolutional architecture in the encoder and decoder networks to enable efficient learning on large datasets and image sizes; (b) it can handle single static frames as opposed to pair of images required in [9]; and (c) it is generative.

## 3 Model

As shown in Figure 1, the basic structure of the Deep Convolutional Inverse Graphics Network (DC-IGN) consists of two parts: an encoder network which captures a distribution over *graphics codes* $Z$ given data $x$ and a decoder network which learns a conditional distribution to produce an approximation $\hat{x}$ given $Z$. $Z$ can be a disentangled representation containing a factored set of latent variables $z_i \in Z$ such as pose, light and shape. This is important

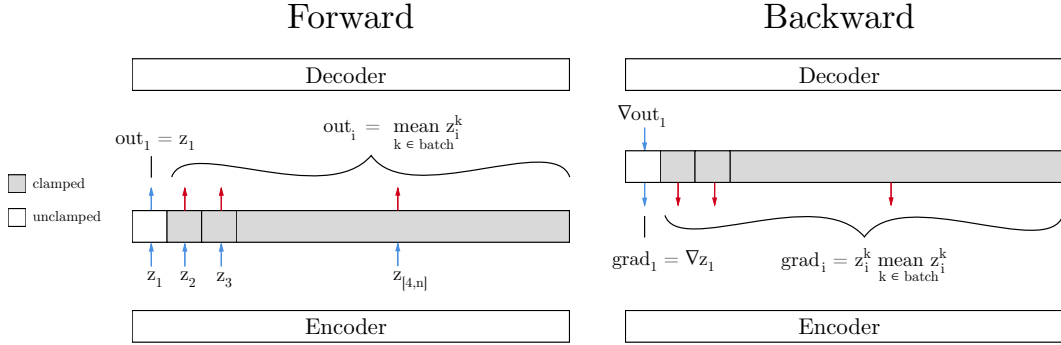

Figure 3: **Training on a minibatch in which only $\phi$, the azimuth angle of the face, changes.** During the forward step, the output from each component $z_i \neq z_1$ of the encoder is altered to be the same for each sample in the batch. This reflects the fact that the generating variables of the image (e.g. the identity of the face) which correspond to the desired values of these latents are unchanged throughout the batch. By holding these outputs constant throughout the batch, the single neuron $z_1$ is forced to explain all the variance within the batch, i.e. the full range of changes to the image caused by changing $\phi$. During the backward step $z_1$ is the only neuron which receives a gradient signal from the attempted reconstruction, and all $z_i \neq z_1$ receive a signal which nudges them to be closer to their respective averages over the batch. During the complete training process, after this batch, another batch is selected at random; it likewise contains variations of only one of $\phi, \alpha, \phi_L, intrinsic$; all neurons which do not correspond to the selected latent are clamped; and the training proceeds.

in learning a meaningful approximation of a 3D graphics engine and helps tease apart the generalization capability of the model with respect to different types of transformations.

Let us denote the encoder output of DC-IGN to be $y_e = encoder(x)$. The encoder output is used to parametrize the variational approximation $Q(z_i|y_e)$, where $Q$ is chosen to be a multivariate normal distribution. There are two reasons for using this parametrization: (1) Gradients of samples with respect to parameters $\theta$ of $Q$ can be easily obtained using the reparametrization trick proposed in [10], and (2) Various statistical shape models trained on 3D scanner data such as faces have the same multivariate normal latent distribution [17]. Given that model parameters $W_e$ connect $y_e$ and $z_i$, the distribution parameters $\theta = (\mu_{z_i}, \Sigma_{z_i})$ and latents $Z$ can then be expressed as:

$$\mu_z = W_e y_e, \ \Sigma_z = \text{diag}(\exp(W_e y_e)) \tag{1}$$
$$\forall i, z_i \sim \mathcal{N}(\mu_{z_i}, \Sigma_{z_i}) \tag{2}$$

We present a novel training procedure which allows networks to be trained to have disentangled and interpretable representations.

### 3.1 Training with Specific Transformations

The main goal of this work is to learn a representation of the data which consists of disentangled and semantically interpretable latent variables. We would like only a small subset of the latent variables to change for sequences of inputs corresponding to real-world events.

One natural choice of target representation for information about scenes is that already designed for use in graphics engines. If we can deconstruct a face image by splitting it into variables for pose, light, and shape, we can trivially represent the same transformations that these variables are used for in graphics applications. Figure 2 depicts the representation which we will attempt to learn.

With this goal in mind, we perform a training procedure which directly targets this definition of disentanglement. We organize our data into mini-batches corresponding to changes in only a single scene variable (azimuth angle, elevation angle, azimuth angle of the light

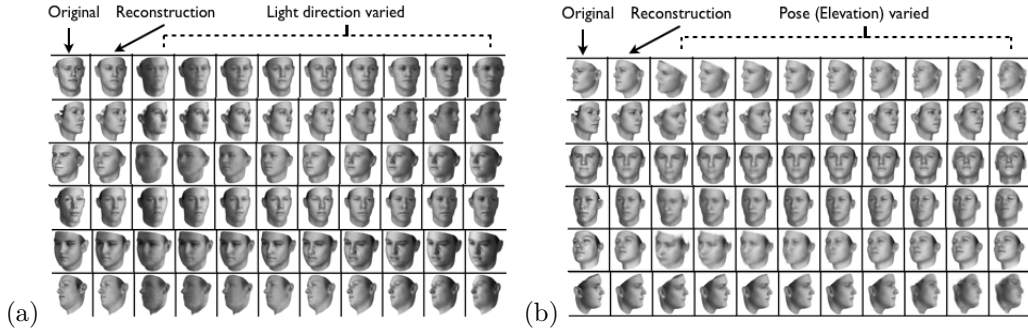

Figure 4: **Manipulating light and elevation variables:** Qualitative results showing the generalization capability of the learned DC-IGN decoder to re-render a single input image with different pose directions. **(a)** We change the latent $z_{light}$ smoothly leaving all 199 other latents unchanged. **(b)** We change the latent $z_{elevation}$ smoothly leaving all 199 other latents unchanged.

source); these are transformations which might occur in the real world. We will term these the *extrinsic* variables, and they are represented by the components $z_{1,2,3}$ of the encoding.

We also generate mini-batches in which the three extrinsic scene variables are held fixed but all other properties of the face change. That is, these batches consist of many different faces under the same viewing conditions and pose. These *intrinsic* properties of the model, which describe identity, shape, expression, etc., are represented by the remainder of the latent variables $z_{[4,200]}$. These mini-batches varying intrinsic properties are interspersed stochastically with those varying the extrinsic properties.

We train this representation using SGVB, but we make some key adjustments to the outputs of the encoder and the gradients which train it. The procedure (Figure 3) is as follows.

1. Select at random a latent variable $z_{train}$ which we wish to correspond to one of {azimuth angle, elevation angle, azimuth of light source, intrinsic properties}.
2. Select at random a mini-batch in which that only that variable changes.
3. Show the network each example in the minibatch and capture its latent representation for that example $z^k$.
4. Calculate the average of those representation vectors over the entire batch.
5. Before putting the encoder's output into the decoder, replace the values $z_i \neq z_{train}$ with their averages over the entire batch. These outputs are "clamped".
6. Calculate reconstruction error and backpropagate as per SGVB in the decoder.
7. Replace the gradients for the latents $z_i \neq z_{train}$ (the clamped neurons) with their difference from the mean (see Section 3.2). The gradient at $z_{train}$ is passed through unchanged.
8. Continue backpropagation through the encoder using the modified gradient.

Since the intrinsic representation is much higher-dimensional than the extrinsic ones, it requires more training. Accordingly we select the type of batch to use in a ratio of about 1:1:1:10, azimuth : elevation : lighting : intrinsic; we arrived at this ratio after extensive testing, and it works well for both of our datasets.

This training procedure works to train both the encoder and decoder to represent certain properties of the data in a specific neuron. By clamping the output of all but one of the neurons, we force the decoder to recreate all the variation in that batch using only the changes in that one neuron's value. By clamping the gradients, we train the encoder to put all the information about the variations in the batch into one output neuron.

This training method leads to networks whose latent variables have a strong *equivariance* with the corresponding generating parameters, as shown in Figure 6. This allows the value

of the true generating parameter (e.g. the true angle of the face) to be trivially extracted from the encoder.

## 3.2  Invariance Targeting

By training with only one transformation at a time, we are encouraging certain neurons to contain specific information; this is equivariance. But we also wish to explicitly *discourage* them from having *other* information; that is, we want them to be invariant to other transformations. Since our mini-batches of training data consist of only one transformation per batch, then this goal corresponds to having all but one of the output neurons of the encoder give the same output for every image in the batch.

To encourage this property of the DC-IGN, we train all the neurons which correspond to the inactive transformations with an error gradient equal to their difference from the mean. It is simplest to think about this gradient as acting on the set of subvectors $z_{inactive}$ from the encoder for each input in the batch. Each of these $z_{inactive}$'s will be pointing to a close-together but not identical point in a high-dimensional space; the invariance training signal will push them all closer together. We don't care where they are; the network can represent the face shown in this batch however it likes. We only care that the network always represents it as still being the same face, no matter which way it's facing. This regularizing force needs to be scaled to be much smaller than the true training signal, otherwise it can overwhelm the reconstruction goal. Empirically, a factor of 1/100 works well.

## 4  Experiments

We trained our model on about 12,000 batches of faces generated from a 3D face model obtained from Paysan *et al.* [17], where each batch consists of 20 faces with random variations on face identity variables (shape/texture), pose, or lighting. We used the *rmsprop* [22] learning algorithm during training and set the meta learning rate equal to 0.0005, the momentum decay to 0.1 and weight decay to 0.01.

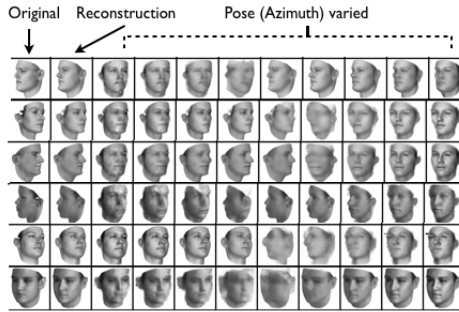

To ensure that these techniques work on other types of data, we also trained networks to perform reconstruction on images of widely varied 3D chairs from many perspectives derived from the Pascal Visual Object Classes dataset as extracted by Aubry *et al.* [16, 1]. This task tests the ability of the DC-IGN to learn a rendering function for a dataset with high variation between the elements of the set; the chairs vary from office chairs to wicker to modern designs, and viewpoints span 360 degrees and two elevations. These networks were trained with the same methods and parameters as the ones above.

Figure 5: **Manipulating azimuth (pose) variables:** Qualitative results showing the generalization capability of the learnt DC-IGN decoder to render original static image with different azimuth (pose) directions. The latent neuron $z_{azimuth}$ is changed to random values but all other latents are clamped.

### 4.1  3D Face Dataset

The decoder network learns an approximate rendering engine as shown in Figures (4,7). Given a static test image, the encoder network produces the latents $Z$ depicting scene variables such as light, pose, shape etc. Similar to an off-the-shelf rendering engine, we can independently control these to generate new images with the decoder. For example, as shown in Figure 7, given the original test image, we can vary the lighting of an image by keeping all the other latents constant and varying $z_{light}$. It is perhaps surprising that the fully-trained decoder network is able to function as a 3D rendering engine.

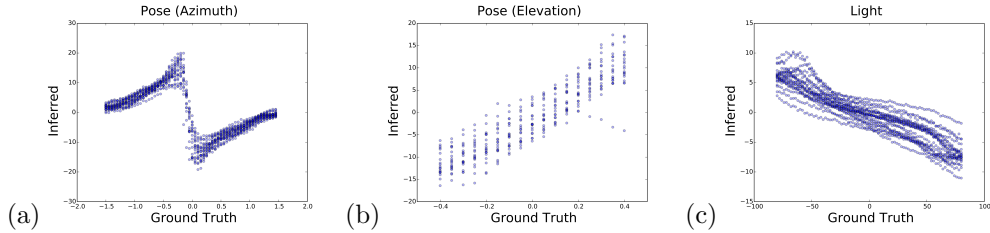

Figure 6: **Generalization of decoder to render images in novel viewpoints and lighting conditions:** We generated several datasets by varying light, azimuth and elevation, and tested the invariance properties of DC-IGN's representation $Z$. We show quantitative performance on three network configurations as described in section 4.1. (a,b,c) All DC-IGN encoder networks reasonably predicts transformations from static test images. Interestingly, as seen in (a), the encoder network seems to have learnt a *switch* node to separately process azimuth on left and right profile side of the face.

We also quantitatively illustrate the network's ability to represent pose and light on a smooth linear manifold as shown in Figure 6, which directly demonstrates our training algorithm's ability to disentangle complex transformations. In these plots, the inferred and ground-truth transformation values are plotted for a random subset of the test set. Interestingly, as shown in Figure 6(a), the encoder network's representation of azimuth has a discontinuity at $0°$ (facing straight forward).

### 4.1.1 Comparison with Entangled Representations

To explore how much of a difference the DC-IGN training procedure makes, we compare the novel-view reconstruction performance of networks with entangled representations (baseline) versus disentangled representations (DC-IGN). The baseline network is identical in every way to the DC-IGN, but was trained with SGVB without using our proposed training procedure. As in Figure 4, we feed each network a single input image, then attempt to use the decoder to re-render this image at different azimuth angles. To do this, we first must figure out which latent of the entangled representation most closely corresponds to the azimuth. This we do rather simply. First, we encode all images in an azimuth-varied batch using the baseline's encoder. Then we calculate the variance of each of the latents over this batch. The latent with the largest variance is then the one most closely associated with the azimuth of the face, and we will call it $z_{azimuth}$. Once that is found, the latent $z_{azimuth}$ is varied for both the models to render a novel view of the face given a single image of that face. Figure 7 shows that explicit disentanglement is critical for novel-view reconstruction.

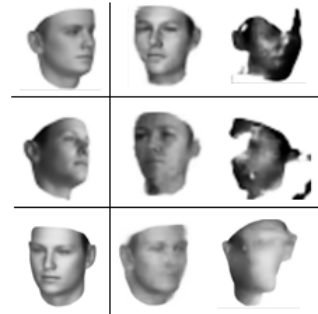

Figure 7: **Entangled versus disentangled representations. First column:** Original images. **Second column:** transformed image using DC-IGN. **Third column:** transformed image using normally-trained network.

### 4.2 Chair Dataset

We performed a similar set of experiments on the 3D chairs dataset described above. This dataset contains still images rendered from 3D CAD models of 1357 different chairs, each model skinned with the photographic texture of the real chair. Each of these models is rendered in 60 different poses; at each of two elevations, there are 30 images taken from 360 degrees around the model. We used approximately 1200 of these chairs in the training set and the remaining 150 in the test set; as such, the networks had never seen the chairs in the test set from any angle, so the tests explore the networks' ability to generalize to arbitrary

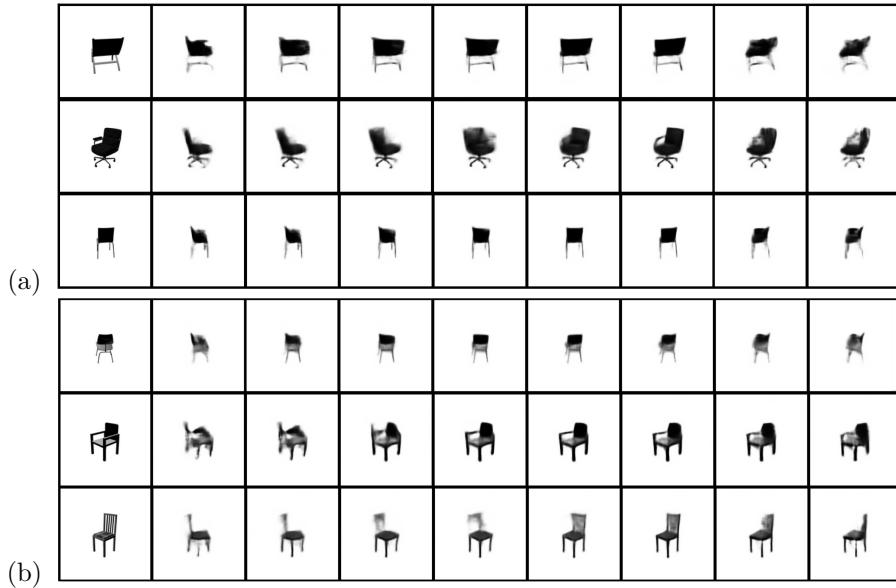

(a)

(b)

Figure 8: **Manipulating rotation:** Each row was generated by encoding the input image (leftmost) with the encoder, then changing the value of a single latent and putting this modified encoding through the decoder. The network has never seen these chairs before at any orientation. **(a)** Some positive examples. Note that the DC-IGN is making a conjecture about any components of the chair it cannot see; in particular, it guesses that the chair in the top row has arms, because it can't see that it doesn't. **(b)** Examples in which the network extrapolates to new viewpoints less accurately.

chairs. We resized the images to $150 \times 150$ pixels and made them grayscale to match our face dataset.

We trained these networks with the azimuth (flat rotation) of the chair as a disentangled variable represented by a single node $z_1$; all other variation between images is undifferentiated and represented by $z_{[2,200]}$. The DC-IGN network succeeded in achieving a mean-squared error (MSE) of reconstruction of $2.7722 \times 10^{-4}$ on the test set. Each image has grayscale values in the range $[0, 1]$ and is $150 \times 150$ pixels.

In Figure 8 we have included examples of the network's ability to re-render previously-unseen chairs at different angles given a single image. For some chairs it is able to render fairly smooth transitions, showing the chair at many intermediate poses, while for others it seems to only capture a sort of "keyframes" representation, only having distinct outputs for a few angles. Interestingly, the task of rotating a chair seen only from one angle requires speculation about unseen components; the chair might have arms, or not; a curved seat or a flat one; etc.

## 5  Discussion

We have shown that it is possible to train a deep convolutional inverse graphics network with a fairly disentangled, interpretable graphics code layer representation from static images. By utilizing a deep convolution and de-convolution architecture within a variational autoencoder formulation, our model can be trained end-to-end using back-propagation on the stochastic variational objective function [10]. We proposed a training procedure to force the network to learn disentangled and interpretable representations. Using 3D face and chair analysis as a working example, we have demonstrated the invariant and equivariant characteristics of the learned representations.

**Acknowledgements:** We thank Thomas Vetter for access to the Basel face model. We are grateful for support from the MIT Center for Brains, Minds, and Machines (CBMM). We also thank Geoffrey Hinton and Ilker Yildrim for helpful feedback and discussions.

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
