[Reviews · NeurIPS 2015]

Submitted by Assigned_Reviewer_1

Although I greatly like this paper, I find the comparison in section 4.1.1 to be a bit of a straw man. I feel that the 'entangled' representation should be driven though (at least) a linear regression, rather than changing just a single activation.
Summary: The paper describes an architecture that can generate novel views of learned 3D objects. The architecture builds on Kingma & Welling, but contributes new ideas including a training procedure that

produces a representation in the autoencoder in which lighting and pose variables are "disentangled".

Submitted by Assigned_Reviewer_2

This is an interesting paper that aims to attach specific meaning to the latent code in a deep convolutional autoencoder. it does this by using a graphics engine to generate images with known pose/lighting etc. and then training the network with data batches chosen to hold variable constant while others vary and then imposing that constraint on the code.

Using synthetic data to have ground truth for many of these extrinsic scene variables makes the technique possible, but also raises the question of how the generally useful the method is. The models learned are only good for generating in the synthetic domain, and it isn't clear if the method could ever be applied to real data (where the g.t. extrinsics aren't available). So while it is a nice idea, its hard to see it having significant long-term impact.

General comments: - The paper is clearly written and covers the relevant work, as far as I can tell. - How vital is the whole variational aspect of the deep AE? Surely the basic idea of the paper would still work just fine if a regular (i.e. point estimate AE) were used. Of course, an extra regularization term on the loss might be needed to replace the KL divergence term in the VAE, but other than that it should work fine. - More generally the main idea in the paper seems agnostic to the precise encoder and decoder used.

- The quality generated samples is ok, but not amazing. The blurring observed is an indication that the model hasn't totally understood the shape. E.g. for faces something like Blantz and Vettter gives much nicer results (admitted for specifically faces).

Probably this is due to the decoder not being a very good generative model. [Making a better decoder seems a more fundamental problem].

- The chairs are indeed a more interesting example.
Summary: Interesting paper which addresses an important problem, namely making the latent representation in deep models interpretable. The central idea of using data with g.t. extrinsics is nice, but not clear how useful it is for real data. The idea is instantiated in a VAE and some decent results are shown.

Submitted by Assigned_Reviewer_3

This paper extends the variational autoencoder framework to learn a graphics code, with latent variables representing specific transformations for the rendering of 3D objects. This is achieved by controlling mini batches and achieves convincing results on two datasets. The method is novel and interesting, with the paper explaining things clearly. However, something that would be interesting to explore further is the baseline entangled representation -- it is far more likely that the azimuth for example will have a distributed representation across latent variables, so perhaps rather than picking a single latent variable with the largest variance to vary, a

linear regressor for azimuth could be trained on top of the latent variables to find the distributed latent representation to vary.

Summary: This paper shows surprising results of a CNN able to emulate a 3D rendering engine, is novel work, and clearly communicated.

Submitted by Assigned_Reviewer_4

Summary: the paper proposes a CNN for learning explicit image representations as an inverse graphics problem.

The image representation has interpretable explicit representations, in particular pose angles and lighting angles, along with implicit representations (texture, appearance).

This is done in an autoencoder framework with reconstruction error.

To make a particular latent dimension focus on one aspect (e.g. azimuth), in each mini-batch, training data is presented with only variations in one aspect, while activations in other nodes are surpressed to a fixed value (the mean value).

Experiments on two datasets showing reconstructions of a 3D object at varying poses and illumination directions.

Quality: The methodology is fine, but some interesting cases are not tested in the experiments. - the pose variables are trained independently.

How well can they be used in conjunction?

For example, changing elevation and azimuth at the same time? - L344-348 - I would have liked to see an actual quantitative evaluation about how well the actual pose could be estimated.

The GT pose and pose variables are in different spaces, correlation between the GT and inferred could be calculated.

Or a simple transformation function from one space to the other could be learned. - Sec 4.1.1 - For the comparison with a standard network, the latent variable with the largest variance is used as the surrogate for azimuth (L359).

It would be interest to also use the linear subspace with the largest variance (i.e., PCA on the latent vector space), since it is possible the azimuth is encoded in multiple latent variables. - no comparisons to other inverse-graphics models [10,17,16,12].

Originality: appears to be original.

The training method for specializing a node for a particular transformation appears novel.

Clarity: - in (1) and (2), why is the mean and variance coupled? i.e., \Sigma = diag(exp(\mu)) - Sec 4.1, it's unclear whether the "static test image" are the same faces as those in the training set. - L376: The error 2.7722e-4 is misleading.

The images in Fig 7 have a lot of white space, which are easy to predict and artifically decreases the error rate.

The error rate should be for only the pixels in the bounding box of the object.

Significance: The paper is interesting, but I'm left to wonder how significant is the result.

What is the end-goal of such a network?

If it is for viewing objects at different viewpoints, it seems limited since the quality of the images is poor and low resolution (looking at Fig 7).

If it is for learning the mapping from image to graphics code (pose parameters), then this is not evaluated in the experiments so it's unclear its advantage over directly predicting the pose parameters with a CNN.

Summary: While the idea is interesting, the results are not very convincing and the signficance is low.

Submitted by Assigned_Reviewer_5

The paper proposes a method to train a variational autoencoder with interpretable latent space representation. In that representation different abstract factors of variation of the inputs are disentangled: a change of only one of the parts of the latent representation corresponds to a change of only one factor of variation of the input. To achieve this, the autoencoder is trained on minibatches of data where only one of the factors of variation changes. The training procedure optimizes a variational lower bound on the log-likelihood of the data with additional terms that correspond to encouraging the whole minibatch to map to the same point in the part of the latent space that corresponds to the factors of variation that are being kept constant.

Quality: The work clearly establishes that their proposed method of learning disentangled representations works by qualitatively assessing the model's ability to change one of the factors of variation in an unseen image. The results look very impressive.

Clarity: The method described is very clear and relevant sources are cited.

Originality: This is an original work --- to reviewer's knowledge clear separation of factors of variation in pictures of 3D objects has not been achieved before in an unsupervised way (with clustering of data according to changes in only one of the factors of variations)

Significance: In reviewer's opinion this is a significant advance towards learning disentangled representations of data in largely unsupervised way.

Small changes: equation (1): is * the matrix product operation? If so, why write it as a star? equation (2): is W_e * y_e in this equation equal to \mu_{z_i}? It seems that a different W_e should be used in this equation. Lines 152-153: it seems what was meant was "Given the model parameters W_e that connect y_e and z_i, the ..." Line 240: "minibatch in which only that variable changes"
Summary: Learning disentangled representations of data has been one of the important goals of machine learning. This paper proposes a method to train variational autoencoders with convolutional layers that learn such representations from unlabeled, but clustered data. The data has to be clustered according to a dimension along which it varies. This framework is very important for representation learning. It can be applied to learning representations that with parts that are invariant or equivariant with respect to group actions, that separate style from content, lighting conditions from identity etc. It is quite obvious that this framework will find future applications in domains where it is easy to generate/obtain data where only one aspect is changing at a time (e.g. multiple speakers say the same thing, multiple pictures are taken of the same object under different lighting conditions etc).

Author Feedback
Author rebuttal: We thank all our reviewers for their thoughtful and encouraging feedback. We answer the questions raised by the reviews below.

(1) What is the end goal or application of this for real data?
While we use synthetic data in our experiments, the only requirement of our approach is the availability of batches of examples in which only one latent variable changes - the learning procedure doesn't need to know the true value of that latent variable. There are several real-world situations which have this property. For example, Reviewer 9 pointed out datasets where multiple speakers say the same words, or where a single object is photographed several times under different affine/lighting conditions- both datasets can be used by our method.

Additionally, our approach shows that it is possible to learn interpretable units. This makes it possible to replace the decoder network with an explicit probabilistic model (by making it differentiable or training it with REINFORCE if non-differentiable). We plan to expand the discussion section in our paper to address this perspective.

(2) The output images are less crisp than more structured models, e.g. the Basel face model by Blanz et al:
Our aim in this paper was to learn an interpretable generative model for a general class of objects, without resorting to special-purpose engineering as in the case of the Basel face model. Therefore we use a general purpose architecture which trades off reconstruction quality with flexibility. That said, modifying the decoder architecture to make it look more like a graphics engine and give it the ability to build richer synthesis models is an important research direction for future work.

(3) Can this model produce results in which more than one of the latents is manually varied at once to produce new images?
We have run these tests and it seems to work pretty well. The results understandably have more noise than changes in only one dimension, but the error seems approximately additive. We will run more thorough tests and include the results in the final version of the paper.

(4) Is the static test image in 4.1 from the training set?
No. All results are shown are using previously-unseen inputs (the test set). We will update the manuscript to make the distinction more clear.

(5) Using a linear combination of entangled latents to re-render instead of just one:
Comparing the disentangled representation to the entangled one was intended to demonstrate that this disentanglement does not just happen by accident. However, we are also quite interested to see how well something like linear regression or PCA can disentangle the representation of the baseline entangled network. We will include results on that test in the final paper.

(6) Is it necessary that this be a variational network, or would any network work?
Variational training allows us to train a generative model which produces coherent samples after learning.

(7) Small textual and notational changes
We thank the reviewers for pointing out typos and notation issues. We will make the suggested corrections and edits in our final draft.